# Frailty Indicator over the Adult Life Cycle as a Predictor of Healthcare Expenditure and Mortality in the Short to Midterm

**DOI:** 10.3390/healthcare12202038

**Published:** 2024-10-15

**Authors:** Carine Milcent

**Affiliations:** 1Paris School of Economics (PSE), 48 Boulevard Jourdan, 75014 Paris, France; carine.milcent@psemail.eu; 2French National Centre for Scientific Research (CNRS), 75014 Paris, France

**Keywords:** healthcare expenditure, frailty indicator, adult life, mortality

## Abstract

Background: Assessing frailty from middle age onward offers valuable insights into predicting healthcare expenditures throughout the life cycle. Objectives: This paper examines the use of physical frailty as an indicator of healthcare demand across all age groups. The originality of this work lies in extending the analysis of frailty indicators beyond the typical focus on individuals under 50 years old to include those in mid-life and older. Methods: For this study, we used a database where frailty was measured in 2012 in a sample of individuals aged 15 to over 90. These individuals were tracked for their healthcare expenditures from 2012 to 2016. Results: Among the sample of 6928 individuals, frailty in 2012 resulted in a statistically significant increase in costs at the 5% level for the population aged 15 to 65. We applied multilevel linear regression models with year fixed effects, controlling for demographic factors, education level, precarity, social dimensions, lifestyle factors (e.g., vegetable consumption), physical activity, emotional well-being, and medical history. A Hausman test was conducted to validate the model choice. For mortality rate analysis, Cox models were used. Conclusions: Our findings demonstrate that physical frailty provides valuable information for understanding its impact on healthcare expenditure. The effect of frailty on mortality is particularly significant for the elderly population. Moreover, frailty is a predictor of healthcare costs not only in older adults but also across the entire life cycle.

## 1. Introduction

State regulation in a health system aims to promote greater equity in access to care. This means ensuring that everyone has access to care when needed by reducing financial barriers. However, understanding the population’s healthcare needs presents a challenge. while the demand for care is often used as a proxy for healthcare needs, it also depends on other factors, such as financial incentives and geographic determinants. Providing free care can lead to opportunistic behavior and overconsumption, so the “demand for care” index must be used with caution. It is crucial for regulators to have an indicator of healthcare needs, but such needs cannot be directly measured. This paper examines how mental and physical frailty, affecting all age groups and not just the elderly, influences the demand for care. One potential approach might involve performing comprehensive health examinations on the entire population simultaneously. However, beyond the enormous expense, the constant evolution of individual health statuses makes it difficult to accurately assess the population’s healthcare needs. The most commonly used health assessment measure is a self-reported one that is predictive of mortality [1] and of healthcare expenditure, but it is subject to biases, which may distort responses and limit its usefulness in healthcare policy implementation [2].

Another branch of indicators is frailty measurement, which assesses an individual’s vulnerability based on physical and mental health problems and their ability to cope with stressors. Frailty is commonly used to identify older adults at risk of disability or declining health [3,4,5,6] and is increasingly recognized as a predictor of healthcare costs. Higher levels of physical frailty are associated with a greater likelihood of poor health, leading to hospitalization, long-term care, or placement in a nursing home. Measuring frailty is linked not only to direct healthcare expenditure—such as the costs of care—but also to indirect costs, such as assistance to frail individuals, health and social services, and caregiver costs. These costs can be considerable and can significantly impact the overall healthcare budget.

Individuals with high levels of physical frailty require more intensive and frequent healthcare, resulting in higher costs. For example, people with multiple chronic conditions, limited mobility, and cognitive impairment require more frequent medical visits, rehabilitation services, and assistance with daily activities [7]. Consequently, frailty indicators serve as valuable tools for anticipating the need for healthcare services. Policymakers use these indicators to allocate resources more effectively, identifying those at highest risk who can benefit from early interventions. The goal is to prevent or delay the need for intensive and costly future care. Thus, frailty indicators provide valuable information for understanding the impact of frailty on healthcare spending and inform decisions about resource allocation and health policy development.

The scientific literature on frailty is extensive and growing, with studies conducted in Japan [8], South Africa [9], Tanzania [10], and China [11]. There is broad consensus that frailty indicators are useful for early detection and thus for enabling the implementation of preventative measures. However, these tools focus primarily on the elderly population. This paper extends the discussion of frailty to include the entire adult life cycle, starting from 15 years of age. This paper investigates the demand for care by examining physical frailty across all ages. To do so, we propose using a physical frailty indicator based on individual fragility, derived from the work of Fried et al. [3]. While the term “frailty index” is commonly associated with the work of Rockwood and Mitnitski [12,13], which requires difficult performance tests, the “frailty phenotype” or physical frailty, as referred to by Fried et al. [2], can be estimated through self-reported questionnaires. However, physical frailty is less precise, as shown by O’Caoimh et al. [14]. Hoogendijk et al. [15] provide an overview of frailty, including distinctions between the frailty index and physical frailty.

This study collected data on physical frailty in 2012 from individuals aged 15 years and older. These individuals were then tracked for their healthcare expenditures from 2012 to 2016. The frailty of this population may be influenced by broader national contexts, including political, economic, and social factors. This paper focuses on the individual factors that impact each person’s frailty index.

The following sections detail the materials and methods used in the study. We then present the study design and data variables, followed by the results. The final section discusses the findings and provides conclusions.

## 2. Materials and Methods

### 2.1. Declaration: Study Design and Participants

This study was conducted in France in the spring of 2012. The Ethics Committee of France approved the study under the number BH_8660 and the declaration of conformity RGPD/CNIL n°2219285.

The French birth rate peaked in 2010 at 12.8 births per 1000 people but steadily declined to 10.6 by 2023. In 2012, the mortality rate was 8.7 per 1000 inhabitants, and the fertility rate was approximately 2. In 2010, the life expectancy was 78 years for males and 84.7 years for females [16]. In Appendix A, Figure A1 presents the current population pyramid. A description of the French health and welfare system financing can be found in Milcent [17,18].

The frailty of the population studied may be influenced by broader national contexts, including its political, economic, and social factors. However, this paper focuses on the individual factors that impact each person’s frailty index.

This study utilizes the French survey ESPS (Santé, Soins et Assurance) conducted by the French National Healthcare Insurance System (NHIS) and the research institute IRDES. This survey is designed to represent the French population in accordance with national statistical methodology (INSEE), with detailed methods described elsewhere [19]. It covers health assessed by the SAH, lifestyle factors, subjective well-being, consumption of medical goods and services, hospitalizations, and individual health status. The main questionnaire collected socio-demographic data on the household of the randomly selected respondents. It was administered via telephone in four calls. For those with unlisted numbers or without a phone, interviewers conducted the questionnaire in person. Additionally, self-administered questionnaires were mailed to the sample population. The NHIS provided information on respondents and their primary care use.

The interview data were merged (1) with the 2012–2016 hospital administrative database containing inpatient healthcare expenses and (2) the 2012–2016 NHIS database containing outpatient healthcare expenses. Data on health expenditures were routinely collected over the four-year period.

To ensure confidentiality, an accredited institute matched these datasets using a unique pseudonymous number contained in each database. All participants provided written informed consent prior to this study.

As inclusion criteria, this analysis required participants to be aged 15 and over with no missing values for physical frailty.

### 2.2. The Level of Frailty as a Driver of Interest

This paper adopts Fried’s method because it offers the potential to prevent the onset of illness. When someone is in a state of pre-frailty, interventions can be made to help them return to a non-frail condition. For example, providing early healthcare support to those exhibiting signs of depression can prevent the development of more serious mental health conditions. This allows for the implementation of a health policy focused on prevention. In 2012, a specific questionnaire addressing frailty was added to the ESPS survey. Fried’s method requires the measurement of five domains: weight loss, exhaustion, grip strength, walking speed, and physical activity [3].

The work of B. Santos-Eggiman et al. [20], based on the first wave of the Survey of Health, Ageing and Retirement in Europe (SHARE), served as a reference for constructing the physical frailty indicator. However, unlike SHARE, which focuses on individuals over 50, this survey includes the entire adult population. Sirven [21] demonstrated that despite the differences between the SHARE and ESPS surveys, there is relative homogeneity between the determinants of frailty. Thus, both surveys are valuable data sources for frailty research.

In the literature, the physical frailty indicator is typically broken down into three categories, based on the number of criteria met: 0 (not frail), 1 or 2 (pre-frail), and 3, 4, or 5 (frail), as in the work of Romero-Ortuno et al. [22] using SHARE data. In this paper, we adopt the same categorization for constructing the physical frailty indicator.

### 2.3. Information Used as Controlled Variables

For the controlled variables, we use the information described in Table 1. Medical history can significantly impact healthcare expenditures. Recent studies [23] have demonstrated how the interaction between pathology and frailty affects healthcare costs.

The social dimension was coded into three categories: single as a reference category (i.e., those not currently having a partner), in a couple but not living together, and cohabiting (i.e., living with a partner, regardless of formal marital status).

The lifestyle dimension included several factors: regular consumption of vegetables or fruit (at least five times per week,) overconsumption of alcohol (more than six glasses per week), smoking or previously smoking regularly, and practicing a sports activity every week.

### 2.4. The Methodology

The aim of this research paper is to investigate how the 3-category physical frailty indicator affects healthcare expenditures across different age groups. Specifically, we examined the impact of frailty on healthcare spending in the “silver age” group (65 years and older) as compared to the younger population (aged 15 to 64). So far, the scientific papers have focused on the aged population [24]. The sample was divided into the silver age group and the younger population group, and econometric models were run for each group. To do so, we analyzed how the physical frailty indicator, measured in year 1, affects healthcare expenditure over a 5-year period.

Additionally, to compare the young population with the mid-life population, we conducted the same analysis, this time dividing the sample at the 50-year-old threshold: the mid-life population (50 years and older) compared to the younger population (aged 15 to 49 years old).

Using data from the two matched administrative databases, we obtained overall hospital and primary healthcare expenditures for each individual, as well as detailed information on the number of days in acute care, long-term care, rehabilitation, and psychiatric care. If applicable, we also recorded the date of death. The dependent variables were as follows:-Acute care hospitalization costs in euros, the number of hospitalizations, and the total length of stays over each year from 2012 to 2016.-Doctor visit expenses (family doctors, specialists excluding optical and dental care) in euros per year, from 2012 to 2016.-Rehabilitation hospitalization, including the number of hospitalizations and total length of stays per year from 2012 to 2016.-Psychiatric hospitalizations, including the number of hospitalizations and the total length of stays per year from 2012 to 2016.-Optical care expenses in euros per year from 2012 to 2016.-Dental care expenses in euros per year from 2012 to 2016.-Overall healthcare expenses (outpatient and inpatient stays) in euros per year from 2012 to 2016.-Date of death, if applicable, from 2012 to 2016.

We used a linear fixed-effects model for the data from 2012 to 2016. For the mortality probability during this period as a dependent variable, we used a Cox model. We conducted a Hausman test to validate the model choice between the fixed-effects model and the random-effects model for both the linear model and the Cox model.

Results tables are presented to compare the impact of the physical frailty indicator on healthcare spending for pre-frail individuals versus non-frail individuals, as well as for frail individuals versus non-frail individuals. The model considered all independent variables listed in the section “Information Used as Controlled Variables”.

## 3. Results

### 3.1. Baseline Characteristics

Table 2 displays preliminary statistics. Half the sample is made up of people under 50. The proportion of women is slightly higher than that of men. One-fifth of the study population is in a state of frailty. The average annual healthcare costs are EUR 1600 before any reimbursement by either compulsory National Health Insurance (NHI) or complementary health insurance. Information on overall expenditure per outpatient and inpatient is given in euros. Detailed information for each care facility is available according to the number of days of the stay.

In Appendix A, Table A1 presents the figures for the independent variables, provided for the entire sample and further categorized by levels of physical frailty. For variable values between 1 and 4, a value of 1 represents a low level, and 4 indicates a high level. Five dimensions are presented: economic, social, lifestyle, emotional, and medical. When an individual is classified as physically frail, both the physical and emotional health indicators show a more deteriorated overall state.

Physical frailty significantly affects healthcare expenditure. Individuals with no frailty (a physical frailty indicator of 0) had average healthcare expenses of approximately EUR 1600. Pre-frail individuals (physical frailty indicator = 1) had average healthcare expenses of EUR 2200. In comparison, frail individuals (physical frailty indicator = 2) had average healthcare expenses of almost EUR 7000. Additionally, the impact varies with age (Figure 1); as individuals get older, their frailty level results in a tenfold increase in average healthcare expenditures.

As expected, frailty levels increase with age (Figure 2). Frailty levels rise from age 15 to 35, stabilize until around age 65, and then enter a new phase of accelerated increase.

Table 3 presents the mortality rate for each frailty level over the observation period. Pre-frailty is associated with a higher mortality rate compared to the non-frail state. Individuals classified as frail, as defined by the physical frailty indicator, exhibit an even higher mortality rate. There is also an increase in mortality within the two years following a pre-frailty assessment compared to those with no frailty. Over the entire observation period, mortality rates continue to rise for frail patients (those with a physical frailty of level 2).

### 3.2. Physical Frailty Indicator as a Driver of Healthcare Expenditures

In France, as shown in Table 4, the total healthcare expenditures for individuals over the age of 65 and those over 50 show no statistically significant increase at the 5% level for individuals classified as pre-frail. However, being frail does result in a significant increase in costs. Furthermore, a comparable result is found over the life cycle. This pattern is consistent across different age groups. Specifically, for the population aged 15 to 50, pre-frailty does not significantly impact expenditures, while being frail leads to notably higher costs. A similar result is observed in the population aged 15 to 65.

The pre-frailty state serves as an early warning of an individual’s declining health, providing an opportunity to revert to a non-frail condition. Therefore, early detection of pre-frailty is crucial to delaying progression to irreversible frailty.

This result is all the more significant as it highlights the minimal increase in out-of-pocket expenditures for pre-frail individuals within the French healthcare system. In contrast, once patients cross the threshold into frailty, expenditure increases for both the healthcare system and the patient.

Looking closely at the composition of healthcare expenditures allows us to better understand how the physical frailty indicator can predict future spending in each category of care.

Table 5 displays inpatient expenditures. Acute care stays are quantified by the generalization of day ambulatory stays for surgery and medical care, ranging from no overnight stays to two days. Long-term care and psychiatry care expenditures are presented in terms of number of days, as the length of stays varies by facility. Some facilities manage patient care through multiple short stays, while others prioritize extended, continuous care without successive returns home.

In both the midlife and “silver age” populations, being frail increases the number of acute care stays at a significance level of 5%. This effect holds true across the entire life cycle. However, the model did not find a significant result for younger individuals. For those over 50, the model found an effect of frailty at an 11% significance level. Additional regression analysis on individuals aged 50 to 65 also found a significant impact of frailty on the number of acute care stays.

Regarding rehabilitation care, the results show that being classified as frail increases cost, whereas being pre-frail does not. Moreover, this effect of frailty on healthcare expenditure remains consistent across all population groups: “silver age”, midlife, as well as young people aged 15 to 50 (and 15 to 65).

Psychiatric care results show that the number of days spent in psychiatric care is actually reduced when someone is identified as frail. Specifically, the physical frailty indicator does not appear to affect psychiatric expenditure for individuals over 50 or over 65 either at the pre-frailty or frailty thresholds. However, for younger populations, being identified as frail has a negative and significant impact on the number of psychiatric stays. This result suggests that for young people classified as frail, the focus is on their physical health rather than their mental health.

Turning to outpatient medical consultations (Table 6), the physical frailty indicator shows a significant 1% increase in healthcare spending on doctor visits. This finding is crucial because it suggests that assessing health needs through the physical frailty indicator is relevant to both older and younger populations. Consequently, this indicator could be expanded to cover the entire lifespan. When combined with other health indicators as a subjective self-assessment of health, it could predict healthcare spending in the near and intermediate future, such as over a five-year span.

For dental and optical outpatient visits, the results show a negative correlation between frailty levels, as defined by the physical frailty indicator, and expenditure on these services. This suggests that patients with deteriorating (pre-frailty) or deteriorated (frailty) health may either reduce or avoid spending on dental and optical care.

### 3.3. Physical Frailty Indicator as a Driver of Mortality

The results, displayed in Table 7, show that at a significance level of 5%, the physical frailty indicator affects individuals’ life cycle. This effect is prevalent in individuals who are identified as frail as well as those who are identified as pre-frail. Therefore, it is suggested that the physical frailty indicator should consider the individual’s life cycle.

However, a more detailed analysis reveals that the frailty indicator has a significant effect of 1% only for frailty and pre-frailty patients over the ages of 50 and 65. For younger patients, the frailty indicator has no effect at the 5% threshold, meaning that it cannot be considered as a predictor of mortality in younger populations. This finding is consistent with the observation of a naturally low mortality rate in the younger population.

### 3.4. Limitations and Sensitivity Analysis

For the sensitivity analysis, models were also run without controlling for chronic diseases, medical history, and emotion indices. The results remained consistent, indicating the robustness of the findings.

The model covers the adult lifespan, starting at age 15, and was tested across different age groups: youth (15–49 years), middle age (50–64 years), and older adults (65 years and over). Despite these variations, the results remained unchanged.

Making a definitive statement about the correlation between healthcare expenditure and the physical frailty indicator based on a single study can be challenging, as numerous factors—such as health status, behaviors, access to healthcare, and responsiveness to interventions—must be considered. However, the physical frailty indicator has proven to be a valuable predictor for future healthcare utilization and costs across the life cycle. Individuals with higher frailty scores tend to require more medical attention, hospitalizations, and long-term care, resulting in increased healthcare expenses over time.

The observation period is from 2012 to 2016. Although the final year of data is not recent, it predates the COVID-19 pandemic by four years, meaning the pandemic’s impact did not influence the results. Nevertheless, the results obtained here suggest that it would be beneficial to replicate this study in a post-pandemic period to assess any potential changes.

One limitation of this study is that frailty was only measured at the beginning of the observation period. While this allows for analysis of the relationship between pre-frailty and healthcare expenditure, it does not consider frailty as a dynamic, evolving state. For example, an individual who is not frail today may become frail in a few years, and the model does not account for this. Additionally, the dimensions controlled in the model may be affected by changes in frailty. For instance, a significant life event such as a divorce may impact an individual’s frailty, thereby affecting social or economic dimensions. These effects are not accounted for in this study.

Another limitation is the set of proxies used to address the demand for care. Although this study uses a broad set of controlled variables, there may be others not included that could better capture this demand. Further research could explore additional indicators to improve the model. However, the wide range of controlled variables and the use of multilevel linear regression and Cox models contributes to the robustness of the results.

Finally, the method of data collection, which includes self-reported survey data and may potentially introduce bias due to gaps in sample coverage from the telephone survey, may introduce possible confounding variables that should be considered when interpreting the results.

## 4. Discussion and Policy Implications

The pre-frailty state indicates declining health but still allows for a return to a non-frail state. Early detection is crucial to prevent progression to frailty. This study explores the relationship between frailty/pre-frailty and healthcare expenditure by comparing costs for frail/pre-frail individuals with those for non-frail individuals. Understanding the potential savings from targeting pre-frail individuals to prevent further decline could guide public health policy. Analyzing the composition of healthcare expenditures helps predict future spending in each category using the physical frailty indicator. Furthermore, this study provides insights into how healthcare expenditure savings vary by age group, highlighting the impact of frailty on healthcare costs across the lifespan, not just in older age or from mid-life onward.

A similar study by Jin et al. [25] focused on older adults, examining whether frailty could account for differences in healthcare expenditure beyond multimorbidity and disability among older Chinese adults. Their findings showed that frailty independently predicted higher healthcare costs in this demographic. These findings highlight the importance of early screening and identification of frailty to reduce healthcare costs for older adults. Our research, using a French database, supports these conclusions across different age groups.

This study underscores the connection between frailty and healthcare expenditure while controlling for various dimensions of individual life. Similarly, Doody et al. [26] conducted a systematic review on frailty among geriatric hospital inpatients, linking it to higher hospitalization rates regardless of demographic, economic, social, emotional, and medical factors. Our study confirms that frailty is an independent predictor of hospital admissions.

It is well documented in the literature that adults aged 50 and older are more susceptible to frailty, leading to increased healthcare needs. Using longitudinal data, Fogg et al. [27] showed that frailty in aging populations is associated with higher use of primary and secondary care services, as well as increased costs. They also noted that individuals with mild and moderate frailty contribute to higher overall costs—a finding consistent with our study, though not statistically significant at the 5% level. Fogg’s study used a 36-deficit electronic Frailty Index (eFI) score from primary care records to determine frailty levels, which was not accessible for our work. Nevertheless, our study confirms increased healthcare spending even for pre-frail individuals, even among younger populations.

For outpatient expenses, extended results are conducted according to Chi et al. [28], Bock et al. [29], and Sirven [30], using French data. While Bock et al. [25] relied on self-reported health service utilization, which could be influenced by recall bias, our study utilized administrative retrospective databases to obtain accurate health service utilization information. Sirven’s [31] study revealed increased health expenses for individuals aged 65 and over as their frailty level rose. This study confirms these findings across various age groups.

This significant finding suggests that the physical frailty indicator is relevant across the lifespan. When combined with other health indicators as a subjective self-assessment health indicator, it could potentially forecast healthcare spending in the near and intermediate future.

This paper presents a groundbreaking finding: individuals aged 50 and under incur higher overall costs when frail. Early intervention for this population could potentially lower service use and costs at a broader level. Our key finding suggests that regulators should take proactive steps to assist the younger population in improving their health through initiatives aimed at preventing frailty. However, this raises ethical questions that should be examined through qualitative studies.

Additionally, this paper reveals that healthcare expenditures for mental health, dental, and eye care decrease with deteriorating (pre-frail) and deteriorated (frail) health, suggesting that these aspects of healthcare are often neglected.

The literature highlights the importance of oral health in frail elderly populations (Van der Putten et al. [32]). A pioneering study by van der Heijden et al. [27,33] linked multidimensional self-reported frailty scales with oral health indices, aiming to identify and anticipate poor oral health and frailty at an early stage. Our study contributes to this growing body of literature by showing that when individuals are frail, oral health is neglected, not only among the elderly but also among younger age groups.

Regarding eye health, a literature review on the relationship between eye diseases and frailty [34] found that regular eye examinations could serve as a preventative measure against ocular diseases linked to frailty. In our study, we observed a negative correlation between eye care expenditure and pre-frailty/frailty, indicating that eye care may be overlooked even for people who can return to “non-frail” health status.

Neglecting oral and eye care may lead to increased disability and higher health- and welfare-related costs.

Frailty is closely linked to mortality, as demonstrated in numerous systematic reviews and meta-analyses across developed and developing countries [35,36]. However, this study’s findings indicate that for younger patients, the frailty indicator does not have an impact at the 5% threshold and cannot be regarded as a mortality predictor in younger populations.

Our study also reveals that younger people classified as frail have a significant negative effect on the number of psychiatric stays, suggesting that when young people are deemed frail, the emphasis is on their physical health rather than their mental health.

In terms of health policy, targeted public health initiatives need to be developed to address the mental healthcare, oral health, and eye health needs of frail and pre-frail populations, who may still return to a “non-frail” health status.

Many existing healthcare systems are ill-equipped to handle vulnerable patients’ chronic and complex medical requirements. Frailty could serve as an ideal risk stratification paradigm. However, there are concerns that labeling individuals based on their level of frailty could lead to stigmatization and negative expectations. According to Kojima et al. [37], addressing these concerns and adopting a public health approach that includes screening, identification, and treatment of frailty could lead to improved care and healthier aging for the elderly.

Our findings on physical frailty support the idea of complementing subjective health indicators, as suggested by Ruth et al. [38], for later life. Thus, ongoing research is needed to develop frailty indicators less prone to subjectivity bias.

## 5. Conclusions

In this paper, we explored the demand for healthcare by examining physical frailty across all ages. The originality of the approach lies in the inclusion of frailty throughout the life cycle, which is rarely addressed in the existing literature. This study demonstrates that physical frailty is a key determinant of health outcomes and healthcare utilization across the life cycle, providing valuable insights into the impact of frailty on healthcare expenditure. Evidence of a significant relationship between healthcare expenditure and the physical frailty indicator can be established at the 5% significance level. While this relationship is particularly pronounced among elderly people with a high physical frailty indicator and an increased risk of adverse health outcomes, it also extends to young people aged 15 to 50 (or 15 to 65). Although the impact on mortality is more significant for the elderly population, the physical frailty indicator remains a valuable predictor of healthcare utilization and expenditure across the lifespan.

From a policy perspective, this paper highlights the importance of implementing health policies focused on early prevention, using the frailty indicator starting in early adulthood and continuing throughout the life cycle. This approach aims to mitigate the decline in health and its associated healthcare costs, which is a central concern in public health research, particularly in managing healthcare expenditures. Notably, the physical frailty indicator, as one component of an individual’s overall health status, provides valuable information for predicting healthcare utilization and costs. It is also closely linked to other factors, such as access to care and health-related behaviors, and can help inform long-term healthcare planning.

## Figures and Tables

**Figure 1 healthcare-12-02038-f001:**
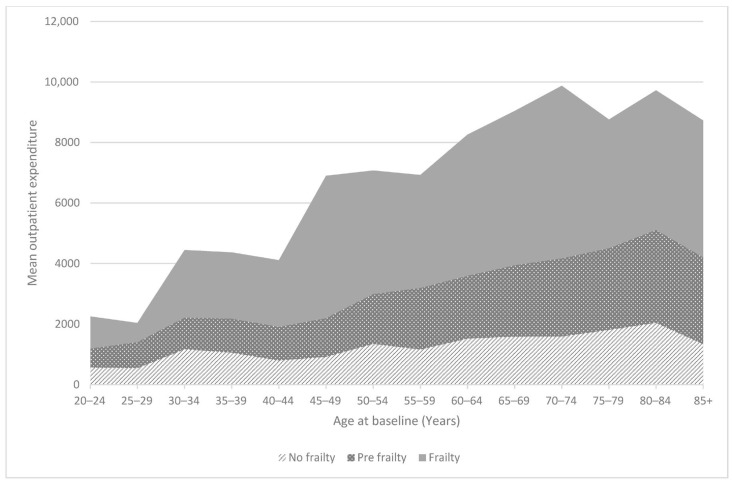
Means of expenditure in 2012.

**Figure 2 healthcare-12-02038-f002:**
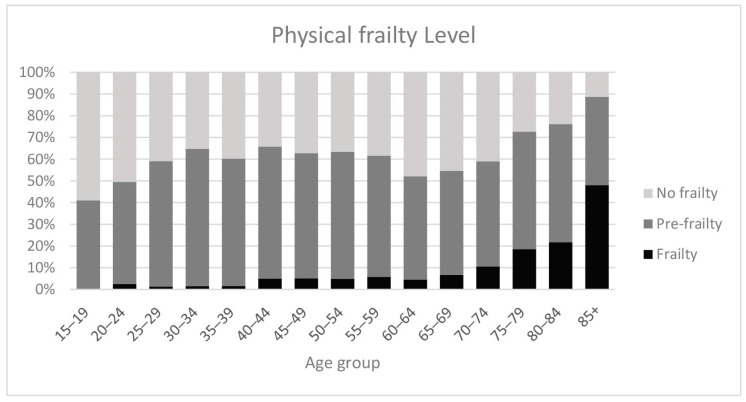
Physical frailty indicator, 2012.

**Table 1 healthcare-12-02038-t001:** Information used as controlled variables.

Variable Group	Variable	Category
Demographic		
	Age	15–29: 0 = no, 1 = yes
		30–49: 0 = no, 1 = yes
		50–64: 0 = no, 1 = yes
		65–79: 0 = no, 1 = yes
		80 and over: 0 = no, 1 = yes
	Sex	0 = male, 1 = female
Medical history		
	Asthma	0 = no, 1 = yes
	Bronchitis	0 = no, 1 = yes
	Infarction disorder	0 = no, 1 = yes
	Coronary disease	0 = no, 1 = yes
	Arterial hypertension	0 = no, 1 = yes
	Stroke	0 = no, 1 = yes
	Osteoarthritis/arthrosis	0 = no, 1 = yes
	Chronic low back pain	0 = no, 1 = yes
	Cervical disorder	0 = no, 1 = yes
	Diabetes	0 = no, 1 = yes
	Allergies	0 = no, 1 = yes
	Cirrhosis	0 = no, 1 = yes
	Urinary disorder	0 = no, 1 = yes
	Depression	0 = no, 1 = yes
	Chronic diseases	0 = no, 1 = yes
	No medical history issue	0 = no, 1 = yes
Economic dimension		
	Primary	0 = no, 1 = yes
	Middle school	0 = no, 1 = yes
	General secondary education	0 = no, 1 = yes
	Tertiary/university education	0 = no, 1 = yes
	Precarity	0 = no, 1 = yes
Social dimension		
	Single	0 = no, 1 = yes
	Couple but not living with the partner	0 = no, 1 = yes
	Cohabiting	0 = no, 1 = yes
Lifestyle dimension		
	Eating vegetables or fruit regularly	0 = no, 1 = yes
	Consumption of alcohol	0 = no, 1 = yes
	Smoking	0 = no, 1 = yes
	Sports	0 = no, 1 = yes
Emotional status		
	Happiness	5-point scale from never to always
	Sadness	5-point scale from never to always
	Calmness	5-point scale from never to always
	Hopelessness	5-point scale from never to always
	Nervousness	5-point scale from never to always

Note: Education level was categorized into four groups: primary, middle school, general secondary education, and tertiary/university education (reference group) to reflect the primary divisions in the French education system.

**Table 2 healthcare-12-02038-t002:** Overall statistics.

In 2012			
Variable		Obs.	Mean
Age			
	15–29	1125	16.24%
	30–49	2359	34.05%
	50–64	1830	26.41%
	65–79	1198	17.29%
	80 and over	407	5.87%
Sex			
	Female	3695	53.33%
Frailty level			
	Non-frail	5504	79.45%
	Pre-frail	1252	18.07%
	Frail	172	2.48%
Total expenditures in EUR		6928	1600.44
Hospital expenditures in days			
	Acute care	6928	0.26
	Rehabilitation care	6928	0.31
	Psychiatric care	6928	0.28
Visit doctors/specialists in EUR			
	Family doctor expenditures	6928	1538.30
	Optical care expenditures	6928	108.27
	Dental care expenditures	6928	166.49

**Table 3 healthcare-12-02038-t003:** Mortality rate and physical frailty indicator.

	Mortality Rate			
Physical Frailty Indicator	2012	2013	2014	2015
No frailty (0)	0.00%	0.22%	0.18%	0.44%
Pre-frailty (1)	0.13%	0.56%	0.86%	0.65%
Frailty (2)	2.42%	3.16%	4.43%	4.88%

**Table 4 healthcare-12-02038-t004:** In- and outpatient expenditures over the period 2012–2016: effect of frailty levels in 2012.

	Population Studied		Physical Frailty Code	Coef.	Std. Err.	z	*p* > |z|	[95% Conf. Interval]	
**Total expenditure**									
Over the lifespan (over 15)									
		Pre-frailty	1	100.810	56.572	1.780	0.075	−10.069	211.690
		Frailty	2	600.355	126.097	4.760	0.000	353.209	847.501
Before midlife (from 15 to 50)									
		Pre-frailty	1	−12.784	56.473	−0.230	0.821	−123.470	97.902
		Frailty	2	367.594	174.760	2.100	0.035	25.071	710.117
From midlife (over 50)									
		Pre-frailty	1	188.398	103.046	1.830	0.068	−13.568	390.364
		Frailty	2	691.007	190.164	3.630	0.000	318.291	1063.722
Before the “silver age” (from 15 to 65)									
		Pre-frailty	1	86.760	55.266	1.570	0.116	−21.559	195.079
		Frailty	2	712.869	151.799	4.700	0.000	415.348	1010.389
From the “silver age” (over 65)									
		Pre-frailty	1	203.538	177.083	1.150	0.250	−143.538	550.613
		Frailty	2	504.526	276.994	1.820	0.069	−38.372	1047.424

Note: The model controlled for the following dimensions: Economic dimension: level of education (primary school, middle school, high school), precarity. Social dimension: cohabitation, couple. Lifestyle dimension: eating vegetables, alcohol consumption, smoking, sports activity. Emotional status: happiness, sadness, calmness, hopelessness, nervousness. Medical history: asthma, bronchitis, infarction, coronary disease, arterial hypertension, stroke, osteoarthritis/arthrosis, chronic low back pain, cervical disorder, diabetes, allergies, cirrhosis, urinary disorder, depression, chronic diseases. Demographic dimension: each 5-year group age, sex. Time fixed effects (FEs) is preferred, with a *p*-value significant (<0.05) for each model.

**Table 5 healthcare-12-02038-t005:** Inpatient expenditure over the period 2012–2016: effect of frailty levels in 2012.

Inpatient	Population Studied		Physical Frailty Code	Coef.	Std. Err.	z	*p* > |z|	[95% Conf. Interval]	
**Acute care hospitalization**									
In number of stays									
	Over the life (over 15)								
		Pre-frailty	1	0.025	0.015	1.720	0.085	−0.003	0.054
		Frailty	2	0.097	0.033	2.960	0.003	0.033	0.161
	Before the midlife (from 15 to 50)								
		Pre-frailty	1	0.018	0.018	0.980	0.329	−0.018	0.053
		Frailty	2	0.045	0.056	0.800	0.426	−0.066	0.155
	From the midlife (over 50)								
		Pre-frailty	1	0.022	0.024	0.940	0.347	−0.024	0.068
		Frailty	2	0.111	0.044	2.550	0.011	0.026	0.196
	Before the silver age (from 15 to 65)								
		Pre-frailty	1	0.020	0.016	1.260	0.209	−0.011	0.050
		Frailty	2	0.069	0.043	1.610	0.108	−0.015	0.153
	From the silver age (over 65)								
		Pre-frailty	1	0.055	0.039	1.410	0.158	−0.021	0.130
		Frailty	2	0.149	0.060	2.470	0.013	0.031	0.268
**Rehabilitation care**									
In number of days									
	Over the life (over 15)								
		Pre-frailty	1	0.125	0.102	1.220	0.223	−0.076	0.326
		Frailty	2	1.108	0.228	4.850	0.000	0.660	1.555
	Before the mid-life (from 15 to 50)								
		Pre-frailty	1	0.023	0.070	0.320	0.748	−0.115	0.160
		Frailty	2	0.481	0.217	2.210	0.027	0.055	0.907
	From the mid-life (over 50)								
		Pre-frailty	1	0.207	0.207	1.000	0.316	−0.198	0.612
		Frailty	2	1.336	0.381	3.500	0.000	0.589	2.084
	Before the silver age (from 15 to 65)								
		Pre-frailty	1	−0.008	0.066	−0.120	0.905	−0.137	0.121
		Frailty	2	0.894	0.181	4.950	0.000	0.540	1.248
	From the silver age (over 65)								
		Pre-frailty	1	0.596	0.439	1.360	0.175	−0.265	1.457
		Frailty	2	1.234	0.687	1.800	0.073	−0.113	2.581
**Psychiatry care**									
In number of days									
	Over the life (over 15)								
		Pre-frailty	1	−0.188	0.105	−1.790	0.073	−0.393	0.017
		Frailty	2	−0.577	0.233	−2.480	0.013	−1.034	−0.120
	Before the midlife (from 15 to 50)								
		Pre-frailty	1	−0.216	0.174	−1.240	0.215	−0.558	0.125
		Frailty	2	−1.230	0.540	−2.280	0.023	−2.288	−0.172
	From the midlife (over 50)								
		Pre-frailty	1	−0.130	0.101	−1.290	0.196	−0.328	0.067
		Frailty	2	−0.181	0.186	−0.970	0.330	−0.545	0.183
	Before the silver age (from 15 to 65)								
		Pre-frailty	1	−0.177	0.120	−1.470	0.141	−0.413	0.059
		Frailty	2	−0.707	0.330	−2.140	0.032	−1.354	−0.060
	From the silver age (over 65)								
		Pre-frailty	1	−0.262	0.210	−1.250	0.211	−0.673	0.148
		Frailty	2	−0.383	0.328	−1.170	0.242	−1.025	0.259

Note: The model controlled for the following dimensions: Economic dimension: level of education (primary school, middle school, high school), precarity. Social dimension: cohabitation, couple. Lifestyle dimension: eating vegetables, alcohol consumption, smoking, sports activity. Emotional status: happiness, sadness, calmness, hopelessness, nervousness. Medical history: asthma, bronchitis, infarction, coronary disease, arterial hypertension, stroke, osteoarthritis/arthrosis, chronic low back pain, cervical disorder, diabetes, allergies, cirrhosis, urinary disorder, depression, chronic diseases. Demographic dimension: each 5-year group age, sex. Time fixed effects (FEs) is preferred, with a *p*-value significant (<0.05) for each model.

**Table 6 healthcare-12-02038-t006:** Outpatient expenditure over the period 2012–2016: effect of frailty level in 2012.

Inpatient	Population Studied	Physical Frailty Code		Coef.	Std. Err.	z	*p* > |z|	[95% Conf. Interval]	
**Outpatient visit**									
	Over the life (over 15)								
		Pre-frailty	1	50.429	61.214	0.820	0.410	−69.549	170.407
		Frailty	2	895.266	136.425	6.560	0.000	627.877	1162.654
	Before midlife (from 15 to 50)								
		Pre-frailty	1	−62.845	64.454	−0.980	0.330	−189.172	63.481
		Frailty	2	833.717	199.119	4.190	0.000	443.451	1223.983
	From midlife (over 50)								
		Pre-frailty	1	169.042	109.279	1.550	0.122	−45.141	383.224
		Frailty	2	914.869	202.164	4.530	0.000	518.635	1311.103
	Before the “silver age” (from 15 to 65)								
		Pre-frailty	1	0.659	61.199	0.010	0.991	−119.290	120.608
		Frailty	2	1148.701	167.912	6.840	0.000	819.600	1477.803
	From the “silver age” (over 65)								
		Pre-frailty	1	329.845	182.965	1.800	0.071	−28.760	688.450
		Frailty	2	842.575	287.927	2.930	0.003	278.248	1406.902
**Eye doctor outpatient visit**									
	Over the lifespan (over 15)								
		Pre-frailty	1	−9.647	3.707	−2.600	0.009	−16.912	−2.382
		Frailty	2	−28.584	8.246	−3.470	0.001	−44.746	−12.421
	Before midlife (from 15 to 50)								
		Pre-frailty	1	−7.728	4.718	−1.640	0.101	−16.974	1.518
		Frailty	2	−2.635	14.524	−0.180	0.856	−31.101	25.831
	From midlife (over 50)								
		Pre-frailty	1	−9.826	5.915	−1.660	0.097	−21.420	1.768
		Frailty	2	−33.793	10.968	−3.080	0.002	−55.290	−12.296
	Before the “silver age” (from 15 to 65)								
		Pre-frailty	1	−8.457	4.226	−2.000	0.045	−16.740	−0.175
		Frailty	2	−27.262	11.565	−2.360	0.018	−49.929	−4.596
	From the “silver age” (over 65)								
		Pre-frailty	1	−17.967	7.801	−2.300	0.021	−33.256	−2.678
		Frailty	2	−39.702	12.345	−3.220	0.001	−63.898	−15.507
**Dental doctor outpatient visit**									
	Over the life (over 15)								
		Pre-frailty	1	−13.018	8.730	−1.490	0.136	−30.128	4.092
		Frailty	2	−67.019	19.417	−3.450	0.001	−105.075	−28.963
	Before midlife (from 15 to 50)								
		Pre-frailty	1	−16.987	9.522	−1.780	0.074	−35.649	1.676
		Frailty	2	−29.336	29.283	−1.000	0.316	−86.730	28.057
	From midlife (over 50)								
		Pre-frailty	1	−2.170	15.386	−0.140	0.888	−32.327	27.986
		Frailty	2	−71.201	28.527	−2.500	0.013	−127.112	−15.289
	Before the “silver age” (from 15 to 65)								
		Pre-frailty	1	−13.645	9.218	−1.480	0.139	−31.713	4.423
		Frailty	2	−50.622	25.207	−2.010	0.045	−100.026	−1.218
	From the “silver age” (over 65)								
		Pre-frailty	1	−9.359	23.483	−0.400	0.690	−55.384	36.666
		Frailty	2	−81.806	37.107	−2.200	0.027	−154.535	−9.077

Note: The model controlled for the following dimensions: Economic dimension: level of education (primary school, middle school, high school), precarity. Social dimension: cohabitation, couple. Lifestyle dimension: eating vegetables, alcohol consumption, smoking, sports activity. Emotional status: happiness, sadness, calmness, hopelessness, nervousness. Medical history: asthma, bronchitis, infarction, coronary disease, arterial hypertension, stroke, osteoarthritis/arthrosis, chronic low back pain, cervical disorder, diabetes, allergies, cirrhosis, urinary disorder, depression, chronic diseases. Demographic dimension: each 5-year group age, sex. Time fixed effects (FEs) is preferred, with a *p*-value significant (<0.05) for each model.

**Table 7 healthcare-12-02038-t007:** Death hazard ratio over the period 2012–2016: frailty level effect in 2012.

Death:Hazard Ratio								
Population Studied	Physical Frailty Code		Haz. Ratio	Std. Err.	z	*p* > |z|	[95% Conf. Interval]	
Over the life (over 15)								
	Pre-frailty	1	2.293	0.558	3.410	0.001	1.422	3.696
	Frailty	2	5.957	1.805	5.890	0.000	3.289	10.788
Before the midlife (from 15 to 50)								
	Pre-frailty	1	0.115	0.130	−1.910	0.056	0.012	1.058
	Frailty	2	0.190	0.260	−1.220	0.224	0.013	2.768
From the midlife (over 50)								
	Pre-frailty	1	2.876	0.758	4.010	0.000	1.715	4.822
	Frailty	2	7.728	2.486	6.360	0.000	4.114	14.516
Before the silver age (from 15 to 65)								
	Pre-frailty	1	1.215	0.460	0.510	0.608	0.578	2.550
	Frailty	2	2.484	2.114	1.070	0.285	0.469	13.168
From the silver age (over 65)								
	Pre-frailty	1	3.302	1.077	3.660	0.000	1.742	6.259
	Frailty	2	8.564	3.192	5.760	0.000	4.125	17.782

Note: The model controlled for the following dimensions: Economic dimension: level of education (primary school, middle school, high school), precarity. Social dimension: cohabitation, couple. Lifestyle dimension: eating vegetables, alcohol consumption, smoking, sports activity. Emotional status: happiness, sadness, calmness, hopelessness, nervousness. Medical history: asthma, bronchitis, infarction, coronary disease, arterial hypertension, stroke, osteoarthritis/arthrosis, chronic low back pain, cervical disorder, diabetes, allergies, cirrhosis, urinary disorder, depression, chronic diseases. Demographic dimension: each 5-year group age, sex. Time fixed effects (FEs) is preferred, with a *p*-value significant (<0.05) for each model.

## Data Availability

Data protection standards and assurances made as part of the informed consent procedure of ESPS preclude the publication of the source data in publicly available repositories. However, individual data access may be granted within a framework of scientific cooperation. The access policy and procedures are available online: https://www.health-data-hub.fr/, accessed on 1 October 2024.

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
