# Peer review of "Frailty Indicator over the Adult Life Cycle as a Predictor of Healthcare Expenditure and Mortality in the Short to Midterm"

_healthcare, 2024, doi:10.3390/healthcare12202038_

Round 1

Reviewer 1 Report (New Reviewer)

Comments and Suggestions for Authors

1)     Authors should not use the words such as I and We in a scientific paper.

2)     The study has some spelling errors. For instance, mortality was written in small letters in the keywords.

3)     The introduction starts with the research question that should not have been done. Instead, authors should provide the theoretical framework first by providing extensive discussions and state of the art in the field.

4)     All papers should be written using scientific language, avoiding words such as imagine, etc. Therefore, extensive grammar and structural revisions are needed.  

5)     The introduction section is far from providing a related theoretical background for the study. Instead, it provides limited information on the former literature.

6)     Authors should establish a former literature review section and provide an extensive literature review and research gap.

7)     The primary motivation of the study was not successfully uncovered.

8)     The study uses several font colors, which is not required. Besides, there are multiple font styles used in the study. Therefore, the paper is unguardedly prepared.

9)     There are some abbreviations that were not explained explicitly. What are GPs?

10)  Authors should avoid using assertive words in the study. For instance, they stated, “Psychiatric care results suggest that frailty can have a surprising effect.” Such a statement requires extensive theoretical explanations that provide pro and con theoretical arguments.

11)  The Conclusion section repeats the already provided information and lacks concrete policy proposals based on the empirical findings.

Comments on the Quality of English Language

Extensive editing of English language required

Author Response

Comments 1: Authors should not use words such as I and We in a scientific paper

Response 1: I changed the "I" into "we". I also limited the number of "we"

Comments 2: The study has some spelling errors. For instance, mortality was written in small letters in the keywords.

Response 2: The paper has been reviewed for editing by Laura Malone, our editing support staff.

Comments 3: The introduction starts with the research question that should not have been done. Instead, authors should provide the theoretical framework first by providing extensive discussions and state of the art in the field.

Response 3: We changed the introduction according to this suggestion.

Comments 4: All papers should be written using scientific language, avoiding words such as imagine, etc. Therefore, extensive grammar and structural revisions are needed.  

Response 4: The paper has been reviewed for scientific language editing by Laura Malone, our editing support. For instance, the sentence using "imagine" has been changed by "One might consider performing a comprehensive health examination on the entire population simultaneously."

Comments 5:  The introduction section is far from providing a related theoretical background for the study. Instead, it provides limited information on the former literature.

Response 5: The scientific literature on frailty is extensive and growing. The paper proposes the related theoretical background in the following paragraphs: "The scientific literature on frailty is extensive and growing, with studies conducted in Japan ([6]), South Africa ([7]), Tanzania ([8]), and China ([9]). The consensus is that frailty indicators are useful for detecting frailty at an earlier stage and thus putting in place preventative measures. However, these tools focus on the elderly population. This paper considers frailty across the adult life cycle. starting from 15 years of age.

This paper addresses the question of the demand for care by examining physical frailty across all ages. To do so, we propose using a physical frailty indicator based on individual fragility, derived from the work of Fried et al. ([2]). The term "frailty index" is commonly associated with the work of Rockwood and Mitnitski ([10], [11]), which requires difficult performance tests to determine. By contrast, the "frailty phenotype" or physical frailty, as referred to by Fried et al. ([2]), can be estimated through self-reported questionnaire items. However, physical frailty is less precise, as shown by O’Caoimh et al. ([12]). Hoogendijk et al. ([13]) proposed an overview of frailty and the differences between the frailty index and physical frailty.

Comments 6:   Authors should establish a former literature review section and provide an extensive literature review and research gap.

Response 6: I thank the reviewer for this remark. In this paper, the scientific literature on frailty is growing with research based on data from various countries, such as Japan ([6]), South Africa ([7]), Tanzania ([8]), and China ([9]). However, there is a lack of scientific literature on frailty across the adult life cycle, starting from 15 years of age addresses the question of the demand for care by examining physical frailty across all ages.

Comments 7: The primary motivation of the study was not successfully uncovered.

Response 7: the primary motivation is to address the question of the demand for care by examining physical frailty across all ages. It is done with indicators of the demand for care. We here use indicators of demand for care available on the database. Some other indicators can be studied using other databases. We added this limitation in the section "3.4. Limitation and Sensitivity Analysis". We wrote the following paragraph "Another limitation of this study is the set of proxies used to address the question of demand for care. In addition to the variables available here, there may be other variables that can also serve as proxies for the demand for care. Further research could be conducted to expand the set of indicators for demand for care used in this paper."

Comments 8: The study uses several font colors, which is not required. Besides, there are multiple font styles used in the study. Therefore, the paper is unguardedly prepared.

Response 8: We do not use font colors in this paper anymore.

Comments 9: There are some abbreviations that were not explained explicitly. What are GPs?

Response 9: We added explanation to abbreviations and we changed "GPs" by "family doctor"

Comments 10:   Authors should avoid using assertive words in the study. For instance, they stated, “Psychiatric care results suggest that frailty can have a surprising effect.” Such a statement requires extensive theoretical explanations that provide pro and con theoretical arguments.

Response 10: We changed this statement by "Psychiatric care results shows that the number of days spent in psychiatry is actually reduced when someone is identified as frail."

Comments 11: The Conclusion section repeats the already provided information and lacks concrete policy proposals based on the empirical findings.

Response 11: following these remarks, the conclusion has been changed as following "In this paper, we addressed the question of the demand for care by examining physical frailty across all ages. This study has shown that physical frailty can be a key determinant of health outcomes and healthcare utilization all over the life cycle, providing valuable insights into the impact of frailty on healthcare expenditure. Evidence of a significant relationship between healthcare expenditure and the physical frailty indicator can be established at the 5% significance level. While this relationship is particularly relevant for elderly people with a high physical frailty indicator and an increased risk of adverse health outcomes, it also extends to young people aged 15 to 50 (respectively, 15 to 65). While the impact on mortality is more significant for the elderly population, the physical frailty indicator remains a valuable predictor of healthcare utilization and expenditure across the lifespan.

 In terms of policy proposals, this paper highlights the importance of implementing a frailty indicator that begins in early adulthood and continues throughout the life cycle of individuals. This approach aims to prevent the decline in health status and its associated consequences in terms of healthcare expenditure."

Reviewer 2 Report (New Reviewer)

Comments and Suggestions for Authors

- Abstract: reword so that you do not use the first person "I"

- The introduction and first few paragraphs should directly address the need for the frailty index expansion to younger populations. Please condense your historical gap references in the first 2-3 paragraphs to a few sentences.

- Define frailty and frailty index in the first or second paragraph.

-  Re: "The introduction of state regulation in a health system aims to promote greater equity of access to care". Please clarify how this impacts care equity, because state regulations are not something new.

- reword to remove the word "I" in: In this paper, I propose using a physical frailty indicator based on 78 individual frailty, derived from the work of Fried..."

- Appendix A: It is not clear what the Mean 23,2% indicates for the Mean of primary school. Perhaps frequency and percentage of the Economic Dimension category? Please add table notation to clarify this for all in Table A1.

- Methods: Move this to the first sentence in Method 2.1 and combine it with the existing first sentence: The baseline survey was conducted in the spring of 2012. E.g. the study was conducted in France in the spring of 2012.

- This should be the secont sentence in Methods 2.1: The Ethics Committee of France approved the study under the number 130 BH_8660 and the declaration of conformity RGPD/CNIL n°2219285.

- Move this sentence to the end of the Introduction as it references the entire paper: The frailty of the population studied may be influenced by the overall 103 context of the country, including its political, economic, and social situations. 104 This paper focuses on the individual factors that impact the frailty index of each 105 individual.

-Re: Methods 2.2: what actions would change pre-fraility to non-frail? Please clarify this association with examples.

_ Table 1: check age category endpoints. For example, some 30 years of age can be put into the first two categories.

- Table2: top line should extend across the whole table

-  Table 5: Add closing bracket: [95% Conf.

- Limitations: add possible confounding variables; self-reported survey data, gaps in sample coverage due to telephone survey method

Comments on the Quality of English Language

The paper was well written with only a minor edits needed for English quality.

Author Response

Comments 1: Abstract: reword so that you do not use the first person "I"

Response 1: We followed the suggestion of the reviewer. The abstract has been changed accordingly.

Comments 2: - The introduction and first few paragraphs should directly address the need for the frailty index expansion to younger populations. Please condense your historical gap references in the first 2-3 paragraphs to a few sentences.

Response 2: Following this suggestion, the first 2-3 paragraphs have been reduced into a single paragraph "One might consider performing a comprehensive health examination on the entire population simultaneously. Beyond the enormous expense, the constant evolution of health status adds to the difficulty of understanding the population's health needs. The most commonly used health assessment measure is a self-reported one that is predictive of mortality [1] and of health care expenditure but it is subject to biases, which can distort responses and limit their use in health policy implementation."

Comments 3: Define frailty and frailty index in the first or second paragraph.

Response 3: Owing to Comment 2, the notion of frailty now appears in paragraph 3, much earlier than in the previous version of the paper.

Comments 4: - reword to remove the word "I" in: In this paper, I propose using a physical frailty indicator based on 78 individual frailty, derived from the work of Fried..."

Response 4: It has been done

Comments 5: - Appendix A: It is not clear what the Mean 23,2% indicates for the Mean of primary school. Perhaps frequency and percentage of the Economic Dimension category? Please add table notation to clarify this for all in Table A1.

Response 5: Appendix A, Table A1 has been changed as suggested.

Comments 6: - Methods: Move this to the first sentence in Method 2.1 and combine it with the existing first sentence: The baseline survey was conducted in the spring of 2012. E.g. the study was conducted in France in the spring of 2012.

Response 6: It has been done accordingly.

Comments 7: - This should be the secont sentence in Methods 2.1: The Ethics Committee of France approved the study under the number 130 BH_8660 and the declaration of conformity RGPD/CNIL n°2219285.

Response 7: It has been done accordingly.

Comments 8: - Move this sentence to the end of the Introduction as it references the entire paper: The frailty of the population studied may be influenced by the overall 103 context of the country, including its political, economic, and social situations. 104 This paper focuses on the individual factors that impact the frailty index of each 105 individual.

Response 8:  It has been done accordingly.

Comments 9: -Re: Methods 2.2: what actions would change pre-fraility to non-frail? Please clarify this association with examples.

Response 9: We changed this paragraph as following "

This paper selected Fried's method because it can help prevent people from falling ill. When someone is in a state of pre-frailty, it is possible to take action to help them regain a non-frail health status. For example, providing early healthcare support for individuals showing signs of depression to prevent mental health conditions. A health policy focused on prevention can then be implemented."

Comments 10: _ Table 1: check age category endpoints. For example, some 30 years of age can be put into the first two categories.

Response 10: It has been done accordingly.

Comments 11: - Table2: top line should extend across the whole table

Response 11: It has been done accordingly.

Comments 12: Table 5: Add closing bracket: [95% Conf.

Response 12: It has been done accordingly.

Comments 13: Limitations: add possible confounding variables; self-reported survey data, gaps in sample coverage due to telephone survey method

Response 13: A paragraph has been added in the Conclusion section "Furthermore, the method of data collection, which includes self-reported survey data but also bias due to gaps in sample coverage resulting from the telephone survey method, may introduce possible confounding variables."

Comments 14: The paper was well written with only a minor edits needed for English quality.

Response 14: the paper has been corrected for editing by L. Malone, our editing support staff.

Reviewer 3 Report (New Reviewer)

Comments and Suggestions for Authors

Frailty Indicator Over the Adult Life as a Predictor of Healthcare Expenditure and Mortality in the Short to Midterm

This article addresses the important issue of frailty in the care needs index by examining physical frailty across age groups. This is an original approach, as frailty indices are typically calculated for people younger than 50 years, compared with middle-aged and older people. Considering frailty in middle age and across the life course can provide a powerful indicator of health expenditure. However, the article contains several shortcomings:

1.    The article lacks a clearly stated research hypothesis.
2.    Please move chapter Limitation after chapter conclusion.
3.    A strength of the article is the research methodology. The authors use a database in which frailty in 2012 was measured in a sample of individuals aged 15 to over 90 years, and these individuals were then tracked for healthcare expenditure between 2012 and 2016. In a sample of 6,928 individuals, it was found that being frail in 2012 resulted in a statistically significant increase in costs at the 5% level for the population aged 15 to 65 years. Multilevel linear regression models with fixed annual effects were run, controlling for demographic factors, education level, economic insecurity, social aspects, lifestyle factors (such as eating vegetables), sporting activity, emotional status and medical history. Cox models were used to analyse mortality rates.
4.    The Results  and Conclusions are also well written. The authors highlight that physical frailty can be a key indicator of health care demand, which has important implications for health policy planning and resource allocation. The use of frailty as an indicator in different age groups allows for a more accurate prediction of future health expenditure and may lead to better management of health resources.
The article is a valuable contribution to public health research, particularly in the context of managing health expenditures. The use of a wide range of controlled variables and multilevel linear regression and Cox models increases the robustness of the results. The originality of the approach lies in the inclusion of frailty throughout the life cycle, which is rarely seen in the literature.

Author Response

Comments 1: The article lacks a clearly stated research hypothesis.

Response 1: The research hypothesis states that we should consider the life cycle in the relationship between frailty and demand for care. The conclusion is affirmative.

Comments 2: Please move chapter Limitation after chapter conclusion.

Response 2: Based on the comments of the 4 other reviewers, it is not typical for reviewers to move the Limitations chapter after the Conclusion chapter.

Comments 3: A strength of the article is the research methodology. [...]

Response 3: We thank the reviewer for these comments.

Comments 4: The Results and Conclusions are also well written [...]. 

Response 4: We thank the reviewer for these comments. In the conclusion, we added some of the reviewer's comments as following "In terms of policy proposals, this paper highlights the importance of implementing a health policy focusing on prevention based on a frailty indicator that begins in early adulthood and continues throughout the life cycle of individuals. This approach aims to prevent the decline in health status and its associated consequences regarding healthcare expenditure. This is central in public health research, particularly in the context of managing health expenditures."

Reviewer 4 Report (New Reviewer)

Comments and Suggestions for Authors

Literature review is missing (except those references in the introduction). The author should present the methodology and results more clearly. Variables in the econometric analysis should be justified accordingly and based on previous literature, in case new variables are included a reason should be present. Fixed effects are used, but no Hausmann test is provided. In stats table some more comments on data are needed. Some typos throughout the text were detected. In Table 1 a yes/no for alcohol consumption is missing; Table 2 (overall stats), the header "mean" does not apply to age groups. The paper is not fluid at all (except introduction) and information provided need to be organized properly. Note that only 20% of all references are recent (2020-2024).

Comments on the Quality of English Language

In general, English is fine; however, editing is needed.

Author Response

Comments 1: Literature review is missing (except those references in the introduction). The author should present the methodology and results more clearly. Variables in the econometric analysis should be justified accordingly and based on previous literature, in case new variables are included a reason should be present. 

Response 1: We have included more than 10 references in Section 4, "Discussion and Policy Implication." Additionally, In order to comply with your suggestion, we have incorporated recent publications in Section 2.2, where the variables used are presented, and in Section 2.4, "The Methodology," where the methodology comparison is explained. When inserting a recent scientific paper, our focus was on justifying the controlled for historical medical variables used in this particular paper. 

[19] Álvarez-Bustos A, Rodríguez-Sánchez B, Carnicero-Carreño JA, Sepúlveda-Loyola W, Garcia-Garcia FJ, Rodríguez-Mañas L. Healthcare cost expenditures associated to frailty and sarcopenia. BMC Geriatr. 2022 Sep 13;22(1):747.

[20]  Lavado À, Serra-Colomer J, Serra-Prat M, Burdoy E, Cabré M. Relationship of frailty status with health resource use and healthcare costs in the population aged 65 and over in Catalonia. Eur J Ageing. 2023 Jun 7;20(1):20

Comments 2: Fixed effects are used, but no Hausmann test is provided. 

Response 2: We conducted a Hausman test to validate the model selection, and we included an additional sentence in the paper.

Comments 3: In stats table some more comments on data are needed. Some typos throughout the text were detected. In Table 1 a yes/no for alcohol consumption is missing; Table 2 (overall stats), the header "mean" does not apply to age groups.

Response 3: We changed Table 1 and Table 2 according to these comments.

Comments 4: The paper is not fluid at all (except introduction) and information provided need to be organized properly. Note that only 20% of all references are recent (2020-2024).

Response 4: About the references, we added more recent scientific papers. About the papers, we changed some paragraphs to make the paper more fluid. However, over the 5 reviewers, remarks on the introduction and changes on the introduction have been required. No remark on the methods and results have been made. One reviewer underlined that "A strength of the article is the research methodology. The authors use a database in which frailty in 2012 was measured in a sample of individuals aged 15 to over 90 years, and these individuals were then tracked for healthcare expenditure between 2012 and 2016. In a sample of 6,928 individuals, it was found that being frail in 2012 resulted in a statistically significant increase in costs at the 5% level for the population aged 15 to 65 years. Multilevel linear regression models with fixed annual effects were run, controlling for demographic factors, education level, economic insecurity, social aspects, lifestyle factors (such as eating vegetables), sporting activity, emotional status and medical history. Cox models were used to analyse mortality rates.
4.    The Results  and Conclusions are also well written. The authors highlight that physical frailty can be a key indicator of health care demand, which has important implications for health policy planning and resource allocation. The use of frailty as an indicator in different age groups allows for a more accurate prediction of future health expenditure and may lead to better management of health resources.
The article is a valuable contribution to public health research, particularly in the context of managing health expenditures. The use of a wide range of controlled variables and multilevel linear regression and Cox models increases the robustness of the results. The originality of the approach lies in the inclusion of frailty throughout the life cycle, which is rarely seen in the literature."

Comments 5: In general, English is fine; however, editing is needed.

Response 5: the paper has been corrected for editing by L. Malone, our editing support staff.

Reviewer 5 Report (New Reviewer)

Comments and Suggestions for Authors

The manuscript analyses healthcare expenditure related to physical frialty indicator in a well focused way and with a wide perspective of research. The topic is relevant, the aim well defined and the results offer food for thought related to overall health financing, sustainability and evolving with population changing needs. Thank you for the opportunity of revising this manuscript, that in my opinion it is worth of publication with minor revisions.

Then, I have some minor suggestions.

Title: if "over the adult life" refers to the broader age range than other studies, it is ok.

Abstract: please revise "our findings". The abstract, and main text, is written in the first person singular.

introduction: extremely well focused. After reading all the paper, and some relevant considerations in the discussion and conclusion, probably it could be useful for an international reader a brief sentence describing French Health and Welfare system financing.

line 49 probably to add a comprehensive reference?

line 65 I suggest referencing

from line 72 typos with parentheses for references, also elsewhere in the text

line 92 to 94 redundant, please remove

line 96 to 102 if available please mention the 2012 data as the study is settled in 2012.

line 121 and 122: better 1) and 2)

please put Figure 2 in the text after having mentioned the figure

line 280 I do not see the significance level notation in the table

line 346 and in the discussion lines from 491 extremely relevant. I suggest the addition of a brief sentence explaining that those missed intervention/controls/treatments means a further increase of disability and then yet health and welfare related costs.

Author Response

Comments 1: Title: if "over the adult life" refers to the broader age range than other studies, it is ok.

Response 1: Indeed.

Comments 2: Abstract: please revise "our findings". The abstract, and main text, is written in the first person singular.

Response 2: Two out of the five reviewers requested that we use "we" instead of "I". We made the change accordingly in the paper.

Comments 3: introduction: extremely well focused. After reading all the paper, and some relevant considerations in the discussion and conclusion, probably it could be useful for an international reader a brief sentence describing French Health and Welfare system financing.

Response 3: Some reviewers have requested that we focus on the problematic area and avoid any other description part in the introduction. However, to address this suggestion, we have added a sentence to indicate where specific details on the French healthcare system can be found.

Etilé F. and Milcent C., “Income-related reporting heterogeneity in self-assessed health: Evidence from France”, Health Economics, 15, 965-981, 2006.

Maresova, P., Javanmardi, E., Barakovic, S. et al. Consequences of chronic diseases and other limitations associated with old age – a scoping review. BMC Public Health 19, 1431 (2019). 

Milcent (2024a) Book "Economie de la santé et des systèmes de santé", ed. Ellipses.

Milcent (2024b) Three Conflicting Objectives for a Single Tool: A case study on French health policy for hospital payments. Hal Working paper, 2024.

Comments 4: 

  • line 49 probably to add a comprehensive reference?
  • line 65 I suggest referencing
  • from line 72 typos with parentheses for references, also elsewhere in the text
  • line 92 to 94 redundant, please remove
  • line 96 to 102 if available please mention the 2012 data as the study is settled in 2012.
  • line 121 and 122: better 1) and 2)

Response 4: All these suggestions have been followed

Comments 5: please put Figure 2 in the text after having mentioned the figure

Response 5: We changed accordingly

Comments 6: line 280 I do not see the significance level notation in the table

Response 6: We agree. It has been removed.

Comments 7: line 346 and in the discussion lines from 491 extremely relevant. I suggest the addition of a brief sentence explaining that those missed intervention/controls/treatments means a further increase of disability and then yet health and welfare related costs.

Response 7: as suggested, the following sentence has been added "Those missed intervention, controls or treatments for oral and eye health means a further increase of disability and then yet health and welfare related costs."

Round 2

Reviewer 1 Report (New Reviewer)

Comments and Suggestions for Authors

The authors have done some changes on the manuscript; however, the revisions do not meet my former queries.

Comments on the Quality of English Language

Minor editing of English language required.

Author Response

Dear reviewer

I totally agree with the request for information about the Hausman Test. I modified the text accordingly.    In the Material and Method section: At the end of part 2, I included the following sentence: We used a linear fixed-effect model for the data from 2012 to 2016. For the mortality probability during this period as a dependent variable, we used a Cox model. We conducted a Hausman test to validate the model choice between the fixed-effects model and the random-effects model for both the linear model and the Cox model.   In the Results section, For each Table, we added the following note: the Fixed Effects (FE) is preferred, with a p-value significant (<0.05) for each model.   About the abstract, it was changed by

Abstract: This paper examines the use of physical frailty as an indicator of healthcare demand across all age groups. The originality of this work lies in extending the analysis of frailty indicators beyond the typical focus on individuals under 50 years old to include those in mid-life and older. Assessing frailty from middle age onward offers valuable insights into predicting healthcare expenditures throughout the life cycle

For this study, we use a database where frailty was measured in 2012 in a sample of individuals aged 15 to over 90. These individuals were tracked for their healthcare expenditures from 2012 to 2016. Among the sample of 6,928 individuals, frailty in 2012 resulted in a statistically significant increase in costs at the 5% level for the population aged 15 to 65. We applied multilevel linear regression models with year-fixed effects, controlling for demographic factors, education level, precarity, social dimensions, lifestyle factors (e.g. vegetable consumption), physical activity, emotional well-being, and medical history. A Hausman test was conducted to validate the model choice. For mortality rate analysis, Cox models were used.

Our findings demonstrate that physical frailty provides valuable information for understanding its impact on healthcare expenditure. The effect of frailty on mortality is particularly significant for the elderly population. Moreover, frailty is not only a predictor of healthcare costs in older adults but also across the entire life cycle."

The introduction received high satisfaction from the other four. Besides, reviewer 4 did not mention any specific points. 

The "background and include all relevant references" received high satisfaction from the other four reviewers. Besides, reviewer 4 did not mention any specific points. 

The "Is the research design appropriate?" received high satisfaction from the other four reviewers. Besides, reviewer 4 did not mention any specific points.  

The "Are the methods adequately described?" received high satisfaction from the other four reviewers. However, as I mentioned before, I agree that the specific point on the Hausman test needed improvement, which I have addressed. 

The "Are the results clearly presented?" received high satisfaction from the other four reviewers. Besides, reviewer 4 did not mention any specific points. 

The "Are the conclusions supported by the results?"  received high satisfaction from the other four reviewers. Besides, reviewer 4 did not mention any specific points.  

This manuscript is a resubmission of an earlier submission. The following is a list of the peer review reports and author responses from that submission.

Round 1

Reviewer 1 Report

Comments and Suggestions for Authors

Dear author,

Your topic seems relevant because frailty indicators can help us make important decisions considering the population we are working with. The approach is innovative, and it is fertile ground for research. I will leave some comments to improve the manuscript.

Line 5: The school's address is missing.

Lines 11–15: you must indicate the analysis performed and present key quantitative results such as central trend and scattering measures, hypotheses tests, p-values, and other concrete information.

Line 19: I would not recommend using personal pronouns, but if you choose to do so, use "I" instead of "we" because, as far as it seems to me, there is only one author. You could integrate this paragraph within lines 69 and 76, where you present your objective.

Line 35 and the following: MDPI has a specific citation style. There is some openness to sending any style, but knowing about the publisher's style is advantageous. You can find through the following link: https://www.mdpi.com/authors/references

Line 70: instead of writing "work of Fried (Fried et al., 2001)", you can simply write "work of Fried et al. (2001)". In other words, you do not have to repeat the author's name. Apply this logic for other references. Once again, I recommend the MDPI style (https://www.mdpi.com/authors/references).

Lines 77-80 should be part of "material and method."

Sections 1.1 to 2 should be combined as "material and method."

Section 1.1: It is essential to describe the study area. Was it France entirely, selected provinces, or Paris? It is also vital to describe critical demographic variables such as natality, fertility, longevity, and mortality. A table and the population pyramid could be an asset. Other aspects affecting frailty, including political, economic, social, technological, environmental, or legal, are worth mentioning.

Also, at some point, you must discriminate between the inclusion and exclusion criteria.

Line 90: you must cite the document "IRDES: L'enquête sur la santé, les soins et l'assurance maladie _ ESPS" like the others and include it in the reference list.

Section 1.2: You should start by explaining why you chose Fried's method over others.

Section 1.3: Can you summarize all the text you wrote in one or two tables. The way you did it, it is hard to read. For instance, you could have a column of "variable group" where you include demographic, medical history, education, etc.; then, you could have "variable," where you could consist of age, sex, asthma, bronchitis, etc.; finally, one called "categories," where you include the age categories, "0= male, 1= female", "0= no, 1= yes", etc.

Section 2: you should explain why you choose groups containing overlapping people. For instance, the group "50 years and older" includes "65 years and 154 older". How do you distinguish both? Why did you not define groups that excluded each other? I am not really criticizing. You must explain that to the readers because it can be confusing.

Now I noticed that Table 1 presents the age groups well organized. I would suggest the age groups be the same in the method presented in the table.

Table 1 is good, but attention should be paid to the caption. Follow the journal's guidelines. Do it also with the figures.

The overall analysis (results section) is well done. I do not have much criticism about it. The sensitivity analysis (section 5) must be part of the results, not a separate section.

Section 5 (discussion) is where I think there is much work to do. It is very brief and superficial, considering how detailed the results were. What you present in the discussion is good and constructive, but I think you can substantially enrich it.

This is why you should have described the study area in more detail. Besides political aspects, there are economic, social, technological, environmental, and legal aspects. Furthermore, it could have been helpful if you had discussed related findings (particularly in France) from other authors. Did someone perform a study like yours before? What did they find? If there are differences, why? Which more likely corresponds to the truth? Why? Furthermore, you must discuss every result systematically and confront the literature.

For instance, did other authors find the exact relationship between age and frailty? Why? Who is correct? Why? The same must be discussed about sex, medical history, education, etc. After that, you should discuss policies, economic and social impact, etc.

There is room for a very robust and impactful discussion.

If the Editorial team agrees, your paper will benefit from a theoretical framework between the "introduction" and "material and method" sections in which you explain previous surveys, indicators, why some approaches are better than others, and their impact on the policies. It is even better to focus on the literature based on the French population, but do not restrain yourself too much if there are gems from other countries. This will later support your discussion.

Above all, work on your discussion and conclusion methodically because your paper can reference the matter well.

Yours sincerely

Comments on the Quality of English Language

Dear Author,

Consider improving your writing according to the following recommendations:

Line 9: "aged 65 or younger, 50 years." Did you mean "between 50 and 65 years old?"

Line 97: instead of "merged 1)- with", write "merged (1) with". Do it with number 2 also.

Lines 102-104: correct the sentence to "The Ethics Committee of the French approved the study under..."

Line 115: write "SHARE" with capital letters.

Line 120 and others: pay attention to the tense. The dominant tense is preterite in a report because the study was already conducted.

Yours sincerely

Author Response

Dear Reviewer

I really appreciate your recommendations. We tried, based on all the recommendations of the reviewers, to improve and correct the article.

Thank you for your support and help.

Reviewer 2 Report

Comments and Suggestions for Authors

Thanks for a manuscript on an important topic.

However, information on frailty indicator comes from a survey that was administered in 2012 (and now it is 2024) and health care expenditure data comes from 2012-2016. Unfortunately, data that is mostly a decade old is no longer valid for use in any study.

Author Response

Dear reviewer, 

I thank you for considering this paper a a very interesting read; and that you found good to see the utilization of frailty scoring earlier in life as a predicting term. 

The information on the frailty indicator comes from a survey conducted in 2012 and administrative data from 2012 to 2016.vI agree that further studies using more recent data must be conducted to consolidate these initial results.

A paragraph has been added in the 3.4. "Limitation and Sensitivity Analysis" Section.

"The period under observation is from 2012 to 2016. The last year of observation is not recent, which may be considered a drawback. However, this data predates the COVID-19 pandemic by four years, which prevented its impact on the results. Nevertheless, the results obtained here suggest that it would be beneficial to replicate this study during a post-pandemic period."

The paper has been edited y Ms Laura Malone from Philadelphia, US (laura.malone@psemail.eu).

I really thank you for your interest in this work.

Reviewer 3 Report

Comments and Suggestions for Authors

Very interesting read, good to see the utilisation of frailty scoring earlier in life as a predicting term. However, it is important for extensive editing of English language to take place.

Methods: 1.3: Cumbersome read - please consider using a table; similar at Methods: 2

Discussion and conclusion: these are very short and need to be further enhanced to explain how the modelled predictions used can be utilised in the future. It is important however to understand frailty as a continuum - someone is not frail today but may be frail in 10 years time - how's the model taking that into account?

Despite a very promising premise, the discussion is far too small to support the results, and the actual methods are riddled with the fact that there is no limitation aknowledgment.

Comments on the Quality of English Language

It requires extensive work.

Author Response

Dear reviewer, 

I thank you for considering this paper a a very interesting read; and that you found good to see the utilization of frailty scoring earlier in life as a predicting term. 

I really appreciate your recommendations. We tried, based on all the recommendations of the reviewers, to improve and correct the article.

Methods: 1.3: Cumbersome read - please consider using a table; similar at Methods: 2

As suggested, I changed the (previous) part 1.3 accordingly. 

Discussion and conclusion: these are very short and need to be further enhanced to explain how the modelled predictions used can be utilised in the future. It is important however to understand frailty as a continuum - someone is not frail today but may be frail in 10 years time - how's the model taking that into account?

I completely agree with this remark. I added a complementary comment in the limitation section. I also added information to the Discussion and Conclusion Sections

Despite a very promising premise, the discussion is far too small to support the results, and the actual methods are riddled with the fact that there is no limitation aknowledgment.

I included supporting information in the discussion and acknowledged limitations in the Limitation and Sensitivity Analysis section.

I sincerely appreciate your interest in this paper.

Thank you for your support and help.